# Life-Threatening Conditions and Preoperative Complications Associated with Cardiac Neoplasm Do Not Affect Surgical Outcomes or Mortality

**DOI:** 10.3390/jcm13185532

**Published:** 2024-09-18

**Authors:** Kenji Suzuki, Shun-Ichiro Sakamoto, Atsushi Hiromoto, Motohiro Maeda, Takako Yamaguchi, Naoki Yamada, Hitomi Ueda, Takayoshi Matsuyama, Shin-Ichi Osaka, Yosuke Ishii

**Affiliations:** 1Department of Cardiovascular Surgery, Nippon Medical School Musashikosugi Hospital, Kawasaki 211-8533, Japan; 2Department of Nursing, Nippon Medical School Musashikosugi Hospital, Kawasaki 211-8533, Japan; 3Department of Cardiovascular Surgery, Nippon Medical School Hospital, Tokyo 113-8603, Japan

**Keywords:** cardiac neoplasm, cardiac tumor, life threatening, ventricular tachycardia, cerebral infraction

## Abstract

**Background:** Cardiac neoplasms may cause life-threatening symptoms associated with cerebral infarction, ventricular arrhythmias, and heart failure. Emergency surgery or preoperative treatment may be required for these patients. However, no study has reported the surgical outcomes in cases involving cardiac neoplasms with life-threatening complications. The current study investigated the mid- to long-term outcomes of surgery in patients with cardiac neoplasms in life-threatening conditions. **Methods**: This study retrospectively analyzed 36 consecutive patients who underwent resection for cardiac neoplasms with life-threatening cardiovascular, respiratory, and cerebral nervous system complications from January 2000 to December 2022. Their mean age at surgery was 54.9 years. In terms of fatal events, one patient who experienced a ventricular tachycardia storm caused by a left ventricular neoplasm was placed under deep sedation and managed with a ventilator preoperatively. Seven patients who presented with limb motor paralysis and visual defects had cerebral infarction. Two of the seven patients with cerebral infarction received cerebrovascular treatment before cardiac surgery. **Results**: During the follow-up period, cerebral- and cardiovascular-related deaths were not recorded. All postoperative cerebral and cardiovascular complications were new-onset cerebral infarction (n = 2) (with symptoms that improved during the long term). The mean follow-up period was 6.2 years. The 5- and 10-year survival rates of all patients were 89.8% and 78.7%, respectively. There were no significant differences in postoperative prognosis between patients with preoperative cerebral infarctions and those without. **Conclusions**: The long-term surgical outcome of patients with life-threatening symptomatic cardiac neoplasm was good. Thus, preoperative complications did not affect prognosis.

## 1. Introduction

Cardiac neoplasms are classified as primary and metastatic, with the latter being the most common [1,2] with an incidence rate of 0.2–6% [3]. Conversely, primary cardiac neoplasms are rare, with an incidence of 0.1% [4], and 75% of them are benign [5,6]. Further, 72–83% of primary benign neoplasms are myxomas and 1–21% are papillary elastomas [7,8,9]. Primary cardiac neoplasms are typically completely cured with resection and have good outcomes [10,11]. However, these cardiac neoplasms can cause life-threatening complications, such as cerebral infarction (CI) due to tumor embolization, acute heart failure due to restricted blood flow [11,12], and ventricular tachycardia [13,14]. After risk assessment via imaging and other modalities, these patients may require emergency surgery or preoperative treatment [15]. However, only a few reports have evaluated the effects of life-threatening complications on surgical outcomes in patients with cardiac neoplasms.

The current study aimed to investigate surgical mortality and long-term outcomes in patients with life-threatening symptomatic cardiac neoplasms.

## 2. Materials and Methods

### 2.1. Subjects

This study was approved by the Institutional Review Board of Nippon Medical School before data collection. Between January 2000 and December 2022, 72 surgical procedures for primary cardiac neoplasms were performed at our hospital and affiliated institutions. Pericardiotomy only (n = 1), thrombus (n = 1), and vegetation (n = 1) were excluded. In total, 36 patients who underwent surgery presented with life-threatening complications, such as congestive heart failure, hemodynamic disruption caused by ventricular arrhythmias, and thromboembolic events. The other 36 patients were asymptomatic (Figure 1).

The patients’ mean age at surgery was 54.9 ± 20.1 years. Their comorbidities included hypertension (n = 14, 38.9%), dyslipidemia (n = 13, 36.1%), diabetes mellitus (n = 8, 22.2%), and hyperuricemia (n = 8, 22.2%). Approximately 30% of the patients were smokers. The life-threatening conditions observed were acute heart failure (n = 28), lung edema (n = 9), ventricular tachycardia (n = 6), neoplasm incarceration (n = 5), right ventricle outflow tract occlusion (n = 2), supraventricular arrhythmias (n = 2), cerebral artery embolism (n = 7), and coronary artery embolism (n = 1). A patient with coronary artery embolism presented with transient myocardial ischemia with acute chest pain and ST elevation on electrocardiogram. In this case, acute myocardial infarction was suspected, and coronary angiography was performed. However, significant stenosis was not found. The patient was then diagnosed with left atrial myxoma, and temporary coronary embolism was strongly suspected. In terms of preoperative procedures, four of six patients with preoperative ventricular tachycardia underwent radiofrequency catheter ablation. Two of seven patients with preoperative CI received cerebrovascular treatment. One patient had an infarct involving the right cerebellar hemisphere on their head computed tomography scan. The patient underwent a decompression craniotomy 3 days after onset. Another patient had a stroke attributed to left middle cerebral artery occlusion and was managed with thrombus aspiration on the day of onset. One of five patients with a class 4 condition based on the New York Heart Association classification presented with a 70 mm left intra-atrial neoplasm in the mitral valve, leading to circulatory collapse, which was managed with extracorporeal membrane oxygenation preoperatively. One patient with a ventricular tachycardia storm caused by a left ventricular neoplasm was placed under deep sedation and managed with a ventilator preoperatively. Table 1 shows the other preoperative data.

### 2.2. Methods

#### 2.2.1. Morphology and Pathology

Preoperative echocardiography showed neoplasms in the atrial septum in the left atrium (n = 17), posterior wall of the left atrium (n = 3), left atrial side of the mitral valve (n = 1), left ventricular side of the mitral valve (n = 1), aortic side of the aortic valve (n = 2), left ventricular side of the aortic valve (n = 1), left ventricular myocardium (n = 6), atrial septum in the right atrium (n = 2), right atrial side of the tricuspid valve (n = 1), and right ventricular side of the pulmonary valve (n = 2) (Figure 2). The pathological findings were myxomas (n = 21, 58.3%), papillary fibroelastoma (n = 8, 22.2%), fibroma (n = 2, 5.6%), lipoma (n = 2, 5.6%), myxofibrosarcoma (n = 1, 2.8%), lymphoma (n = 1, 2.8%), and hemangioma (n = 1, 2.8%) (Table 2). The maximum long diameter of the neoplasms removed was 38.2 ± 22.5 mm.

#### 2.2.2. Surgical Technique

In total, 21 (58.3%) patients underwent emergent surgery. In all patients, median sternotomy was performed under general anesthesia. Cardiopulmonary bypass was initiated with aortic perfusion and direct bicaval drainage. In one case, inferior vena cava drainage was established via the right femoral vein due to a neoplasm at the atrial septum in the right atrium. All procedures were performed under cardiac arrest. Table 2 shows data on the mean surgical duration, cardiopulmonary bypass time, and cross-clamp time.

In terms of resection methods, endocardial dissection (n = 13), including one using cryoablation and two using electrocautery, was performed. Further, atrial septal resection (n = 8, including 2 with cryoablation); neoplasm resection from the left ventricular myocardium under intraoperative electroanatomic mapping (n = 5) [14,16]; neoplasm amputation with remnant (n = 3, including 2 with cryoablation); valve slicing for neoplasms at the aortic side of the aortic valve (n = 2) and the left ventricular side of the mitral valve (n = 1); and valve resection for neoplasms at the left ventricular side of the aortic valve, followed by aortic valve replacement (n = 1), and at the left atrial side of the mitral valve, followed by mitral valve replacement (n = 1), were conducted. In one case, the neoplasm occupied almost the whole left ventricular wall. Thus, a partial resection with cryoablation was performed. In a case of myxoma in the pulmonary valve, the neoplasm extended to the right and left ventricular outflow tracts. Thus, a partial resection was performed to release the stenosis in the right ventricular outflow tract. Table 2 shows the data on the combined surgical procedures.

#### 2.2.3. Patient Follow-Up

Patients were followed-up during scheduled visits at 1 month and then every 2 months within the first year to evaluate hemodynamics on electrocardiography and chest radiography. After the second year, the patients were followed-up by a family doctor. The medical records of the patients were reviewed to determine the postoperative recurrence of cardiac neoplasm and any morbidities or mortalities. Some patients were followed-up via telephone.

#### 2.2.4. Statistical Analysis

Data are presented as means ± standard deviations or ranges. Kaplan–Meier curves were used to depict patient survival from the surgery. Between-group differences were compared using the Wilcoxon signed-rank test. For all tests, *p*-values < 0.05 were considered statistically significant. All statistical analyses were performed using JMP 14.2.0 (SAS Institute Inc., Cary, NC, USA).

## 3. Results

### 3.1. Surgical Mortality and Morbidity

One patient died on postoperative day 19 due to sepsis caused by aspiration pneumonia. This patient underwent valve resection for papillary fibroelastoma adhering to the mitral valve, and his postoperative hemodynamics were stable and uncomplicated. On postoperative day 15, he developed aspiration pneumonia due to his advanced age (81 years old), which was complicated by sepsis. Two patients with left atrial myxoma developed postoperative cerebral complications. In particular, one patient developed aphasia and right hemiparesis on postoperative day 2, leading to the diagnosis of CI, with residual mild impairment of motor skills in the right hand. The other patient developed left hemiparesis on postoperative day 4, leading to the diagnosis of CI, which improved with endovascular treatment. None of the patients developed postoperative cerebral hemorrhage. In addition, seven patients with preoperative CI did not present with worsening cerebral complications.

### 3.2. Late Follow-Up

Seven patients (papillary fibroelastoma, n = 3; myxoma, n = 2; lymphoma, n = 1; and myxofibrosarcoma, n = 1) presented with mid- to long-term mortality. Two deaths occurred at 1 and 8 months postoperatively due to aspiration pneumonia. One patient with neoplasm recurrence died of respiratory failure at 2 years postoperatively. There were no cardiac- or cerebrovascular-related deaths in either case.

Two patients presented with neoplasm recurrence. Among them, one had partial resection for the pulmonary valve myxoma, which recurred 9 months after surgery. The patient died of respiratory failure 2 years after surgery. The other patient presented with left atrial myxoma, developed a cerebral aneurysm of the right middle cerebral artery 4 years after surgery, and underwent clipping surgery. Pathological examination showed neoplasm infiltration of the myxoma in the aneurysm vessel wall.

The 5- and 10-year survival rates of all patients were 89.8% and 78.7%, respectively (Figure 3). Further, the 5-year survival rate of seven patients with preoperative CI and 29 patients without were 100% and 79.8%, respectively. The mortality rate did not differ between patients with preoperative CI and those without (*p* = 0.189) (Figure 4).

## 4. Discussion

### 4.1. Outcomes of Cardiac Neoplasms with Life-Threatening Complications

The number of reports on surgery for cardiac neoplasms with life-threatening complications is limited, and all studies only evaluated surgical outcomes in patients with asymptomatic neoplasms. In these reports, the treatment indications were heart failure (26.3–37%) and embolic symptoms including CI (19.3–33%), with a surgical mortality rate of 0–3.5% [8,9,17]. By contrast, our study only reported the surgical outcomes of patients with neoplasms in life-threatening conditions. After risk assessment via imaging and other modalities, these patients may require emergency surgery or preoperative treatment [15]. However, a surgical mortality rate of 2.8% was favorable. In the present study, the most common condition was acute heart failure with neoplasm deviation and obstruction. The results showed that heart failure could be treated with hemodynamic stability. In cases of embolism, the timing of treatment after disease onset was appropriate.

### 4.2. Pathogenesis of Cerebral Complications

Myxoma, which is the most common cardiac neoplasm, often presents with heart failure, followed by embolic symptoms. According to previous reports on myxoma, embolism in cases with myxoma is 21–89% CI and 13–29% other embolization [17,18,19], and embolization is more common in patients with echocardiographic mobility and gross papillary [19,20]. This finding is in accordance with that of the current study, which included 19 patients with myxoma in the left atrium. Among these patients, six (31.6%) presented with preoperative cerebral complications. Moreover, six (85.7%) of the seven patients with preoperative cerebral complications had myxoma in the left atrium.

### 4.3. Risks of Cardiac Surgery after the Development of Brain Complications

The presence of cerebral complications may not be a risk factor for deterioration after cardiac surgeries, including cardiac neoplasm removal. Generally, preoperative brain complications are a risk factor for postoperative deterioration because open-heart surgery requires the use of anticoagulants, such as heparin. In terms of infective endocarditis, which can cause cerebral complications and cardiac tumors, the 2016 American Association for Thoracic Surgery consensus guidelines recommend that surgery should be delayed for 1–2 weeks in patients with nonhemorrhagic strokes [21]. However, the presence of preoperative cerebral complications does not contribute to an increased risk of mortality and neurological complications [22,23]. In addition, hemorrhagic CI caused by cardiac neoplasm can be prevented if the embolus is removed early during stroke onset [24,25]. In fact, our study found that patients with preoperative stroke did not present with deterioration after cardiac neoplasm removal. Two of the seven patients with preoperative CI received cerebrovascular intervention before cardiac surgery, which prevented deterioration after the surgery.

### 4.4. Prospects for Future Treatment Options

This study showed that early intervention might improve the prognosis of patients with life-threatening cardiac tumors. Although the timing of surgery has conventionally been at the surgeons’ preference, early intervention may be encouraged in the future as similar evidence accumulates.

### 4.5. Limitations

This was a retrospective study on the prognosis of patients with life-threatening complications. Therefore, unlike a comparative research study that includes patients without life-threatening complications, this study might have been affected by bias. Further, our cohort was extremely small. Hence, the specific differences among pathological diagnoses could not be statistically determined. A multicenter study should be conducted in the future to investigate factors affecting tumor recurrence and prognosis.

## 5. Conclusions

No cerebral- and cardiovascular-related deaths were recorded during the observation period. The mid- to long-term outcomes of patients with preoperative CI did not differ from those of patients without such complications. Hence, aggressive surgical intervention is effective against cardiac neoplasms with life-threatening complications.

## Figures and Tables

**Figure 1 jcm-13-05532-f001:**
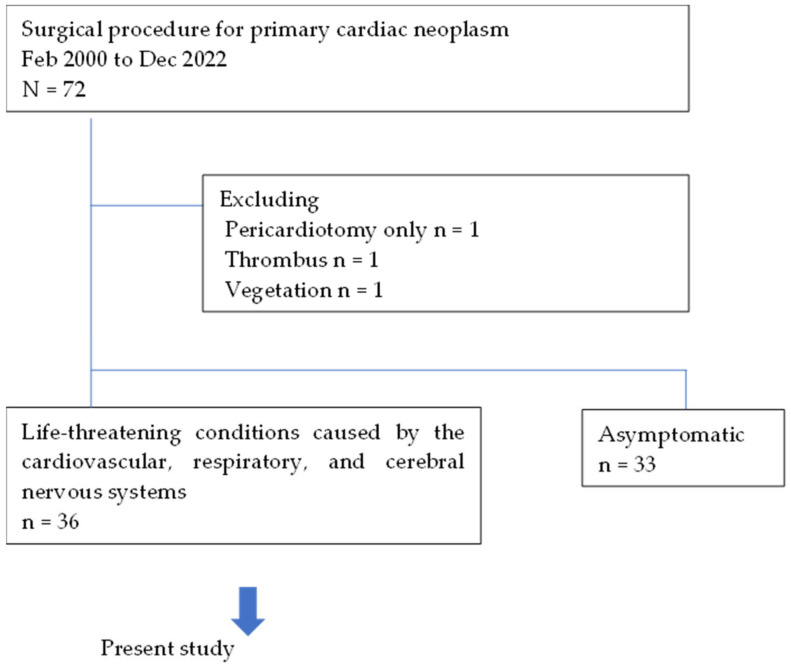
Patient selection.

**Figure 2 jcm-13-05532-f002:**
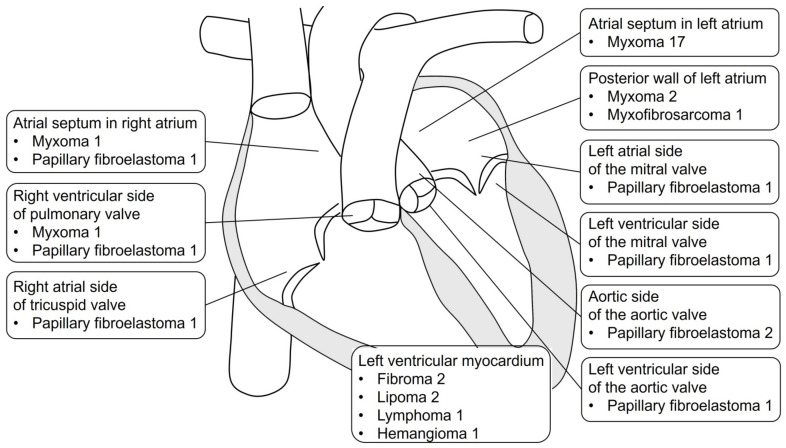
Location and pathological findings of cardiac neoplasm.

**Figure 3 jcm-13-05532-f003:**
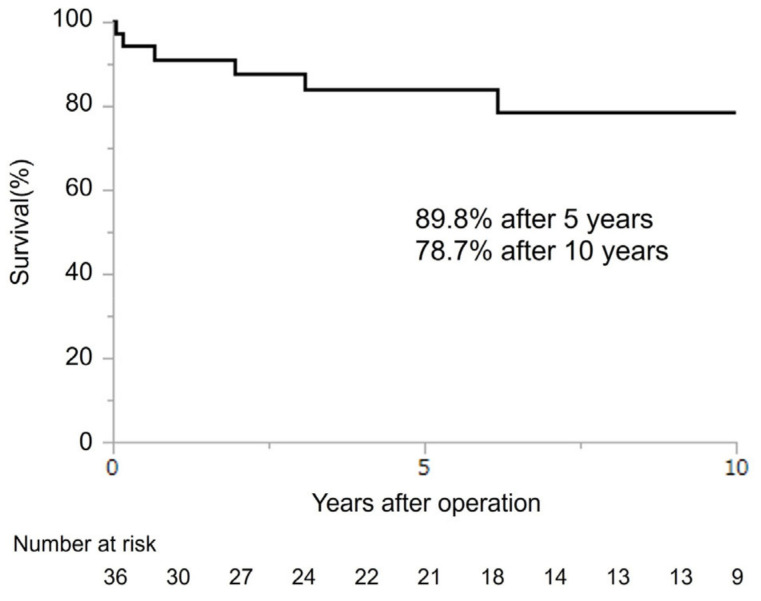
Survival from the time of surgery.

**Figure 4 jcm-13-05532-f004:**
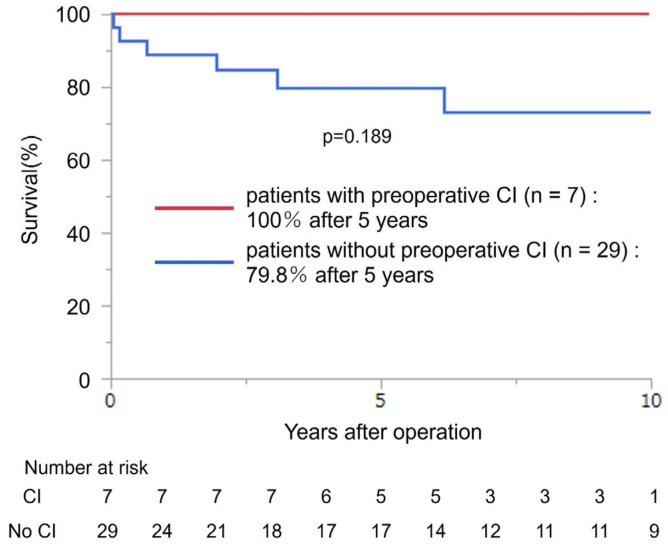
Survival from the time of surgery based on the CI status. CI—cerebral infarction.

**Table 1 jcm-13-05532-t001:** Preoperative status of study participants.

Participants			N = 36 (%)
Age at surgery, years Male sex, n			54.9 ± 20.1 17 (47.2)
Follow-up period, years			6.2 ± 5.3
Comorbidities	Hypertension		14 (38.9)
	Dyslipidemia		13 (36.1)
	Diabetes mellitus		8 (22.2)
	Hyperuricemia		8 (22.2)
Smoking			11 (30.6)
Blood sampling data	Creatinine		0.75 ± 0.32
	C-reactive protein		1.87 ± 2.80
	Prothrombin time:international normalized ratio		1.11 ± 0.12
	Activated partial thromboplastin time		31.8 ± 6.63
	N-terminal pro-brain natriuretic peptide		481.6 ± 630.6
Life-threatening condition	Acute heart failure		28 (77.8)
		Lung edema	9
		Ventricular tachycardia	6
		Neoplasm incarceration	5
		Right ventricular outflow tract occlusion	2
		Supraventricular arrhythmias	2
		Other	4
	Embolism		8 (22.2)
		Cerebral artery	7
		Coronary artery	1
Preoperative procedures	Radiofrequency catheter ablation		4
	Decompression craniotomy		1
	Thrombus aspiration		1
	Extracorporeal membrane oxygenation		1
	Mechanical ventilation		2
Cardiothoracic ratio, %			48.0 ± 6.0
New York Heart Association functional classification, n			I, 19 II, 8 III, 4 IV, 5
Atrial fibrillation, n			2 (5.6)
Echocardiography	Left ventricular end-diastolic diameter, mm		45.0 ± 6.4
	Left ventricular end-systolic diameter, mm		28.6 ± 6.5
	Ejection fraction, %		67.0 ± 10.0
	Left atrial dimension, mm		35.3 ± 7.5
	Aortic valve regurgitation, n		0
	Mitral valve regurgitation, n		>moderate 3 (8.3)
	Tricuspid valve regurgitation, n		>moderate 2 (5.6)

**Table 2 jcm-13-05532-t002:** Surgical technique.

Participants	n = 36 (%)	
Emergency surgery, n	21 (58.3)	
Surgical duration, min	286.9 ± 140.9	
Cardiopulmonary bypass time, min	131.3 ± 81.5	
Cross-clamp time, min	72.4 ± 52.9	
Removal method, n	Endocardial dissection, 13 (36.1) Atrial septal resection, 8 (22.2) Neoplasm resection from the left ventricular myocardium, 5 (13.9) Neoplasm amputation, 3 (8.3) Valve slicing, 3 (8.3) Valve resection, 2 (5.6) Partial resection, 2 (5.6)	Cryoablation, 1; Electrocautery, 1 Cryoablation, 2 Cryoablation, 3 Cryoablation, 2 Cryoablation, 1 Cryoablation, 1
Combined surgery, n	Mitral valve plasty, 2 Tricuspid valve annuloplasty, 1 Coronary artery bypass grafting, 1 Ventricular tachycardia surgery, 3 MAZE, 1	
Pathological diagnosis, n	Myxoma, 21 (58.3) Papillary fibroelastoma, 8 (22.2) Fibroma, 2 (5.6) Lipoma, 2 (5.6) Myxofibrosarcoma, 1 (2.8) Lymphoma, 1 (2.8) Hemangioma, 1 (2.8)	

## Data Availability

The original contributions presented in this study are included in the article, and further inquiries can be directed to the corresponding author.

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
