# Peer review of "Life-Threatening Conditions and Preoperative Complications Associated with Cardiac Neoplasm Do Not Affect Surgical Outcomes or Mortality"

_jcm, 2024, doi:10.3390/jcm13185532_

Round 1

Reviewer 1 Report

Comments and Suggestions for Authors

Kenji Suzuki et al. present an original article with the aim to investigate surgical mortality and long-term outcomes in patients with life-threatening symptomatic cardiac neoplasms. The article is interesting for readers, the methodology of the study is solid with a well-conducted statistical analysis, and the research provides valid conclusions. Tables are well presented but new figures should be added. 

The authors study a topic with poor literature, the main question is investigated, present a manuscript with a good structure and, moreover, conclusions are consistent with the evidence presented. However, key inputs need to be considered to improve the quality and scientific impact of the manuscript and identify gaps in evidence. 

1. I suggest to improve the Introduction by adding the role of cardiovascular imaging in case of cardiac neoplasms, also discussing the role of multimodality imaging. 

2. It is suggested to add the inclusion and exclusion criteria of the study population. 

3. I suggest to add a figure about the study population. 

4. I suggest to improve the Discussion by adding the role of CV imaging in this type of patients with cardiac neoplasms, during risk assessment, before the surgery. It is very important to discuss this point in relation to your results and give possible suggestions and advice in clinical practice to reduce the risk of adverse events.

Please, add this key reference to improve the scientific content of the new sentence: “Perone F, et al. Role of Cardiovascular Imaging in Risk Assessment: Recent Advances, Gaps in Evidence, and Future Directions. J Clin Med. 2023 Aug 26;12(17):5563. doi: 10.3390/jcm12175563”.

 5. What are the gaps in evidence and future directions? Add these points in the manuscript.

 6. It is suggested to add a central figure showing your research findings.

Reviewer 2 Report

Comments and Suggestions for Authors

This study retrospectively analyzed 36 consecutive patients who underwent cardiac tumor resection for life-threatening cardiovascular, respiratory, and cerebral nervous system complications. The authors noted that despite good surgical outcomes for cardiac tumors, the lack of studies on patients with life-threatening complications provided a reasonable motivation for this study. The study found that despite severe complications in patients, no cerebrovascular or cardiovascular-related deaths were recorded postoperatively, and the 5- and 10-year survival rates were 89.8% and 78.7%, respectively. The results suggest that surgical treatment of cardiac tumors in life-threatening situations is safe and effective, predicting that aggressive surgical interventions can lead to a favorable long-term prognosis even in complex clinical settings. The authors provide valuable insights into exploring the outcomes of cardiac tumor surgery in life-threatening situations. Despite the limitations of the small sample size and retrospective design, the study results show a favorable long-term prognosis with clinical implications.

The following issues need to be addressed:

Major issues:

1. In the preface section, the authors describe the epidemiology of cardiac tumors and the clinical presentation of primary and comorbidities and emphasize that previous studies have not evaluated the impact of life-threatening complications on surgical outcomes in patients with cardiac tumors. The reviewer would like to clarify that using only one negative statement to indicate the shortcomings of the previous studies may not be comprehensive or rigorous enough and suggest that a critical analysis of the existing literature be added to highlight the study's innovativeness and necessity.

2. In terms of study design, this study was a retrospective analysis design that analyzed the surgical outcomes of 36 patients between 2000 and 2022. Although retrospective studies can provide valuable data, their inherent risk of bias should be explicitly mentioned in the discussion.

3 It is recommended that a control group be included in the study so that comparisons can be made with patients without life-threatening complications. This will help to more clearly assess the specific impact of life-threatening complications on surgical outcomes.

4. 69 patients underwent primary cardiac tumor surgery in this study. Thirty-six of these patients were determined to have preoperative life-threatening complications, including congestive heart failure, hemodynamic disruption caused by ventricular arrhythmias, and thromboembolic events. This can be considered an inclusion criterion and hopefully will be confirmed by the authors. It is also recommended that the exclusion criteria be detailed in the methodology section so the reader can better understand the sample selection process.

5. An in-depth complication analysis would help understand the surgical risks more comprehensively. Few postoperative complications were mentioned in the study, and the specifics and impact of new complications were not detailed. It is recommended that a detailed description of postoperative complications, including the incidence, type, and severity of new complications, be added to the results section. A standardized complication classification system could better compare outcomes across patients.

6. During the perioperative period, two patients with left atrial mucinous tumors developed cerebral ischemia on postoperative days 2 and 4, and one patient died of sepsis due to aspiration pneumonia on postoperative day 19. The authors are asked to analyze the possible reasons for these conditions in these patients. In particular, whether the deaths of patients with sepsis are associated with certain factors in the perioperative period.

7. Different types of pathology may affect patients' prognoses, and it is suggested that the authors provide an in-depth analysis of the impact of different types of cardiac tumors (e.g., mucinous tumors, papillary fibromas, etc.) on surgical outcomes.

8. The study's limitations, including the small sample size, inherent flaws in the retrospective design, and differences in follow-up time, should be explored in depth in the discussion section. This would clarify the potential impact of these limitations on outcomes and suggest directions for improvement in future studies.

Minor issues:

1. It is recommended that more details be provided in the statistical analysis section, such as the statistical software used and the specific statistical tests performed.

Reviewer 3 Report

Comments and Suggestions for Authors

I have read with great interest the paper by Suzuki et al regarding the safety, mortality and pre-operative complications in relation to cardiac neoplasms. It is a retrospective analysis of 36 consecutive patients who underwent resection for cardiac neoplasms. There is a scarcity of evidence in the literature, therefore even small registries could provide with insightful conclusions. It is quite noteworthy that there were no deaths related to the procedure, in a rather long follow-up period (mean 6.2 years). Moreover, the 5- and 10-year survival rates of all patients were high (89.8% and 78.7%, respectively).

In introduction, epidemiological data regarding cardiac neoplasms is provided, as well as that there are no studies of assessing the effects of life-threatening complications on surgical outcomes in this population.

I found very interesting that you provided a Figure with the location of the neoplasms.

Question 1

Can you provide with a table of demographic and other baseline characteristics of the patinets (smoking?, DM2? Etc – the common baseline characteristics in studies)

Question 2

If baseline characteristics are available are there associations of any of them with mortality rate?

Question 3

In limitations section you mention limitations

1.     there was no clear standard for the timing of surgery in cases of neoplasm frailty

2.     the timing of surgery was based on the surgeon’s preference.

3.     patient cohort was extremely small”

Can you analyse it more ? What do you suggest according to your trial and your limitations, regarding the design of future trials?

Round 2

Reviewer 1 Report

Comments and Suggestions for Authors

The authors have responded satisfactorily to my requests.

Reviewer 2 Report

Comments and Suggestions for Authors

Thank you for your detailed response to the reviewer's comments and the revisions made to the manuscript. After carefully reviewing the revised version, the reviewer believes the authors have satisfactorily addressed the main concerns and issues raised during the review process.

1. The authors have added a critical analysis of the existing literature in the introduction, further emphasizing this study's innovativeness and necessity.

2. In the discussion section, the authors explicitly mention the inherent risk of bias in retrospective study designs and suggest conducting prospective, multicenter studies in the future to further validate the results.

3. Although this study did not include a control group, it is understandable given the rarity of the study population and the small sample size of critically ill patients with cardiac tumors. If conditions permit in the future, it is recommended to include a control group without life-threatening complications to better assess the impact of complications on prognosis.

4. The authors have supplemented the inclusion and exclusion criteria for the study population, giving readers a clearer understanding of the sample selection process.

5. The authors have explained why they did not discuss postoperative complications in detail: the small number of cases made it difficult to summarize statistical patterns. This point is acceptable.

6. For the cases of postoperative cerebral ischemia and sepsis-related death, the authors have provided a causal analysis. The analysis is reasonable considering the deficient number of cases with such severe complications.

7. The authors have disclosed the pathological diagnoses of mid to long-term mortality cases, which helps readers understand the potential impact of different pathological types on prognosis.

8. The discussion section has been supplemented with the study's limitations and indicates that early intervention may improve prognosis, providing a reference for future research directions.

The revised manuscript has greatly enhanced quality and can be considered for acceptance and publication. Once again, I would like to thank the authors for their meticulous work and wish them success in their research endeavors!